# Indolizines and Their Hetero/Benzo Derivatives in Reactions of [8+2] Cycloaddition

**DOI:** 10.3390/molecules26072050

**Published:** 2021-04-03

**Authors:** Eugene V. Babaev, Ivan A. Shadrin

**Affiliations:** 1Chemistry Department, Moscow State University, Leninskie Gory, 1 Str. 3, 119899 Moscow, Russia; i.shadrin666@gmail.com; 2Higher School of Economics, National Research University, 7 Vavilova Str., 117312 Moscow, Russia; 3N. D. Zelinsky Institute of Organic Chemistry, Russian Academy of Sciences, 47 Leninsky Ave., 119991 Moscow, Russia

**Keywords:** indolizine, azaindolizines, benzoindolizines, cyclazine, [8+2] cycloaddition, mechanism, 1,10-cyclizations, catalysts

## Abstract

Peculiarities of [8+2] cycloaddition of acetylenes to indolizines are reviewed. Especially mentioned are indolizines with leaving groups at positions 3 and 5. Cycloaddition to aza- and benzo derivatives are reviewed, as well as 1,10-cyclizations and processes leading to cyclazines where indolizines are intermediates. Mechanistic features (adducts and cycloadducts) and theoretical aspects (one- or two-steps mechanism) are reviewed.

## 1. Introduction

Indolizine (A, Scheme 1) is the simplest heteroaromatic molecule containing both a π-excessive pyrrole and a π-deficient pyridine ring with only one bridgehead nitrogen, the whole system being isomeric with indole and possess pharmaceutical, agrochemical and fluorescent properties [1]. Although indolizine is certainly aromatic, significant alternations of the bond lengths around the ring system were detected by X-ray, NMR and UV spectroscopy and even mass spectrometry in various substituted indolizines. This prompts some tetraene-like character of the compound, in particular its ability to enter into cycloaddition reactions.

Indolizine is usually regarded as the π-excessive heterocycle with the highest electron population of the carbon atom C-3, and the major part of the chemistry of indolizines is simple electrophilic addition and substitution at this position. Cycloaddition of various dienophiles (alkenes and acetylenes) to indolizines leading to derivatives of the cycl[3.2.2]azine (**B**, Scheme 1) is well-known. The mechanism of these reactions is frequently regarded as a rare example of [8+2] cycloaddition, where the tetraene carbon framework of the indolizine bicycle plays the role of an 8 π-electron fragment. In general, this process may be either one-step (concerted) or involve zwitterionic (and even biradical) intermediates, and there is yet no experimental evidence for the nature of the process. 

Cyclazine (**B**) is an interesting 12π-electronic system that breaks the canons of aromaticity. According to X-ray data the structures **B1** and **B2** are not correct (Scheme 2), and the structure rather has a peripheral delocalization of aromatic 10 π-electron system **B3**. Therefore, cyclazine resembles the famous spinning toy **B4** where the handle (which is not rotated) corresponds to nitrogen lone pair. Hence, the structure has a symmetry plane, and this influences the number of positional isomers, say the number of aza- and benzo-derivatives possible for cyclazine (Scheme 3).

Cyclazins and their hetero/benzo derivatives are important from a practical viewpoint. They are fluorescent compounds and have excellent prospects in organic electronics [2,3,4,5,6,7,8]. On the other hand, biological activity was found in cyclazines, and their applications as estrogens and anti-inflammatory compounds are well known [9,10,11].

Cyclazine was first obtained from indolizine by Boekelheide 60 years ago. This author was the first who postulated the [8+2] mechanism. After this time the [8+2] reaction was reviewed several times. The first review by Acheson appeared in 1963 [12] and the next one by Taurins in 1977 [13]. Several reviews were written on the chemistry of cyclazines [1,14,15,16,17,18,19]. In the reviews of Nair and Abhilash [20,21] devoted to [8+2] cycloaddition reactions, only a limited number of indolizine reactions was mentioned.

Therefore, this review may be considered as the first and comprehensive review on the [8+2] cycloaddition reactions between aza/benzo indolizines and acetylenes leading to cyclazines.

## 2. Cyclazines from Indolizines via [8+2] Catalytic Cycloaddition

The first cycloaddition to indolizine (entry **1** in Table 1) was observed by Boekelheide in 1959 [22] (Scheme 4) and later to **2** [23] by using DMAD and heating in toluene in presence of Pd-C. Recently reaction of **3** was reported with MnO_2_ as oxidant [4]. Boekelheide was the first who made the reaction of **4** with non-symmetric alkyne [24] and performed cycloaddition with DMAD to 2-Ph-indolizine **5** [25]; this reaction was repeated recently with **8a**,**b** [26]. 2-Methylindolizine **6** was involved in the reaction with DMAD in 1965 [27], and the same reaction was done for 8-*R*-indolizines **7a**–**c** [28].

Indolizines **9a**–**c** obtained by desulfurization of 2-MeS derivatives were converted to cyclazines [29]. First indolizine **10** substituted by functional groups was involved in cyclization with DMAD in 1974 [30]. Later, this methodology was used to construct cyclophanes by applying 2-MeS-3-CONH_2_ substituted structures **11a**,**b** [31,32]. Later 2-MeS-3-COOR derivatives were hydrolyzed and decarboxylated to **12a**–**c** and converted to cyclazines [29]. A similar methodology was used to construct cyclazine from 2-MeS-7-NMe_2_-indolizine **13b**: desulfurization gave 7-NMe_2_ derivative **13a** and the addition of DMAD gave corresponding cyclazine [2].

Fluoro-substituted indolizines are seldom [33] but 1-fluoro derivatives **14a**,**b** underwent cycloaddition in oxidative condition in presence of Cu(II) salts [34]. One more example to introduce functionality to cyclazine is catalytic cycloaddition of 6,8-diacyl indolizine **17** [35]. In our recent work, we proved that MAC could react with 2-t-Bu indolizine **18** giving cyclazine in open-air [36]. One featured reaction was cycloaddition of indolizine **19** with Mes_2_B-substituted acetylene [37]. 2-Styryl indolizine reacted with DMAD and MAC without a catalyst [38]. Big series of cyclazines (though without the yields) was synthesized and described as estrogens [9,10].

1-Methoxycarbonyl indolizine **15** (Table 1) was converted to cyclazine in order to make cyclophane [39], but this methodology failed. Finally, cyclazines **21c**,**d** were obtained from bis-indolizinylethanes **21a**,**b** (R = Me, t-Bu) and further converted to cyclophanes [39], Scheme 5.

1-(2-Pyridyl)indolizine in reaction with acetylenes **16a**,**b** formed cyclazine [40], Table 1. The product was converted to indolizino-cyclazine **16c**, and DMAD was added for the second time giving bi-cyclazine **16d** (20 h in boiling xylene without catalyst) with a yield of 30%, Scheme 6.

Analogous structure bearing 2-CO_2_Et group **22a** [41] was converted to condensed indolizino-cyclazinone structure **22b**. from which cyclazino-cyclazinone **22c** was obtained with the yield 73% (Pd/C, NO_2_Ph, 20 h), Scheme 7.

A Japanese group made an effort to prepare cyclazines from 1,8-cycloannelated indolizines **23** [42,43,44] containing propylene and butylene bridges, Scheme 8, Table 2. The major finding was the use of DDQ in reaction with dibenzoyl acetylene (DBZA) under extremely mild conditions. Further studies on oxo-derivatives **24** [45] allowed to make cyclazine bearing (in 1 and 8 positions) oxo-propyl group.

Finally, fused indolizines **25** with saturated piperidyl or hexamethyleneimine bridges across 1,8-positions were [46] prepared and involved in cycloaddition with DBZA giving expected cyclazines, Scheme 9. Table 2. However, an attempt to perform similar reaction with DMAD caused cyclazine formation with unsaturated azepine ring.

Novel reaction conditions were found for cycloaddition reaction, so that the role of oxidant was played by O_2_ in presence of Pd(OAc)_2_ [47], Scheme 10. Many 1-alkoxycarbonyl derivatives (**26a**–**l**) were involved in the reaction with acetylenes of the type ArC≡CAr, Table 3.

A range of indolizine **27** smoothly underwent visible-light-induced intermolecular cyclization with internal alkynes with acceptor group to afford cyclazines in good to excellent yields with high regioselectivity [48], Scheme 11, Table 4.

An efficient visible-light-induced intermolecular [8+2] alkenylation–cyclization process was developed for indolizines **28** [49], Scheme 12, Table 5. In this reaction alkene (not alkyne) formed cyclazine derivatives with oxygen as an oxidant via cascade reaction.

Annulations of 1-cyanoIndolizine with unsaturated carboxylic acids **29a**–**f** was observed during the catalysis with Pd(OAc)_2_ [50], via similar cascade reaction Scheme 13, Table 6.

## 3. Non-Catalytic Cycloaddition to 3- or 5-Substituted Indolizines

If a leaving group X is located at position 3 or 5 of indolizine ring, cycloaddition reaction does not require a catalyst/oxidant for dehydrogenation, because the dihydrocyclazine intermediate can lose HX, Scheme 14.

Such groups X can be -OR or -OCOR. -SR, -NR_2_ or -NR-NR_2_, halogen and some others, Scheme 15.

Thus, 3-acyloxy indolizines **30a**–**d** were converted to cyclazines with excellent yield [51]. Tris-1,2,3-(iso-propylthio)indolizine **31** also underwent such cycloaddition [52]. 3-Hydrazine-substituted derivatives **32a**–**c** lost the attaching group forming cyclazines [53]. Quite similarly behaved 5-substituted indolizines. After refluxing in aromatic solvents, 5-OTms indolizines **33a**–**f** [54], 5-morpholyl **34** [55] and 5-bromo derivatives **35** [56] smoothly formed the expected cyclazine structures in the absence of catalyst. 

## 4. Features of Cycloaddition of 3-Cyano Indolizines and Their Benzo Derivatives

3-CN-Indolizines are the structures that looked capable to react with acetylenes without catalyst due to probable loss of HCN from intermediate. In 1980 the Matsumoto group (together with L. Paquet) reported the first reaction of 3-CN-inolizines with DMAD [57], [58]. 3-Cyanindolizine **36a** and its 6,8-dimethyl analog **36b** with DMAD in refluxing toluene gave expected cyclazines, though in presence of Pd-C (Scheme 16, Table 8). The later group of Tominaga converted 2-MeS-derivatives of 3-CN-indolizines-**37a,b** to MeS-cyclazines (again in the presence of the same catalyst) [59] (Scheme 16, Table 8). 

The most dramatic story happened to another adduct of CN-indolizines and DMAD. In 1980 the Matsumoto group found that 7-methyl- and 7-benzyl derivatives gave 1:2 adduct with proposed structure **39a** [58], Scheme 17. Later the same group tested the reaction of 3-CN indolizines **38a**–**g** in the presence and absence of a catalyst [60,61], Table 8. Finally, the structure of the 1:2 adduct formed without the catalyst was proved by X-ray, and it was unexpectedly styryl pyrrole **39b** [60,61], Scheme 17. Different mechanisms of benzene ring formation and E-group migration have been proposed.

Cyano-derivative of benzo[a]indolizine is easily available from pyridinium-dicyanmethylide and dehydrobenzene. Matsumoto first published the results of cycloaddition of dibenzoylacetylene to the structures **40a**–**d** (Scheme 18, Table 9) [62,63]. Again, the reaction required a catalyst. Tominaga group made this cycloaddition **41** with DMAD [64]. Finally, this reaction was tested extensively with various acetylenes **42** [65].

## 5. Cycloaddition to Benzoindolizines: Synthesis of Benzo Derivatives of Cyclazines

Cycloaddition of benzyne (generated differently) to indolizine **43** is the simplest route to benzo derivatives of cyclazine [3], Scheme 19, Table 10. The resulting structures are strongly fluorescent.

Condensed structures from **43r**,**s** with coumarin ring were similarly obtained, Scheme 20 [3].

Another route to the same benzo-skeleton is cycloaddition of alkynes to benzo[a]indolizines. This reaction was studied with acetylenes containing boron substituents, alone **44a**–**c** [37] or together with nitrogen-containing heterocycle on another end of acetylene **45a**–**e** [5], Scheme 21, Table 11. In one experiment **46** benzyne was generated from PhBr; this resulted in dibenzocyclazine was obtained with low yield [66].

Tominaga showed that indolizines **47a**,**b** having annelated benzene ring across the bond C7–C8 underwent [8+2] cycloaddition forming benzo[g]cycl[3.2.2]azines [29,67], Scheme 22.

In another paper [68], he demonstrated a similar reaction of dibenzoindolizine **48** with DMAD leading to dibenzo[a,h]cycl[3.2.2]azine, Scheme 23.

Isomeric indolizines **49a**,**b** annelated across the bond C6–C7 with benzothiophene underwent cycloaddition with DEAD (PhMe/Δ/6h) without any catalyst [69], Scheme 24.

The last example is 1,2,5,6-dibenzocycl[2,2,3]azine obtained with a yield of 54% from dibenzoindolizine and DEAD in presence of Pd-C [70], Scheme 25.

This reaction is featured, firstly, because it was the first cycloaddition in the history of indolizines that even made an influence on Boekelheide. Second, is that the structure of dibenzoindolizine is extremely polyenic (annelation in indolizine appears across two single bonds), and therefore, the process could be better treated as [2+16] rather than [2+8] cycloaddition.

## 6. Cycloadditions Where Indolizines Are Intermediates

There are many examples of cyclazine synthesis where the intermediates are indolizines. First, there are so-called 3 component reactions: picoline and bromoketone in the presence of a base (Chichibabin combination to obtain indolizine) and alkyne. Two examples of such combination were reported in microwave conditions [71,72], Scheme 26, Table 12.

Another example is given by cycloaddition to pyridone **53a** giving cyclazine **53b** [73], Scheme 27. Evidently, intermediates are (partially isolated) indolizine **55e** which is obtained by sequence **55c**–**55d**.

Another example of cyclazine **54b** synthesis from pyridine **54a** with ethyl propiolate via indolizine **54c [74]** is illustrated in Scheme 28. Indolizine **54c** could be isolated.

A similar reaction is between the same pyridine and benzyne [75,76,77] forming dibenzoindolizine, Scheme 29, Table 13.

In brackets—yield of benzoindolizine. A—diphenyliodonium-2-carboxylate monohydrate 200 °C; B—anthranilic acid and isopentyl nitrite in refluxing chloroform-acetone; C—6-cyanobenzo[a]indolizine diphenyliodonium-2-carboxylatem monohydrate in DME 200 °C 3 h.

Interesting multistep reaction starting from pyridine **57a** and finishing with cyclazine **57b** with the yields 15–70% was observed independently by Acheson and Pohjala [51,78,79,80,81,82], Scheme 30. The mechanism of this process included Perkin reaction and intermediate formation of indolizine skeleton **57c**.

## 7. Cycloaddition to Azacyclazines and Their Benzo-Derivatives

The first cycloaddition to aza-analogs of indolizine was observed by Boekelheide [83] in the reaction of imidazo[1,2-a]pyridine **58** with DMAD in presence of Pd-C, Scheme 31 and Table 14. It was also shown that 6-azaindolizine **59** [84] (but not 7-aza-derivative [25]) can be involved in a similar process. Soon it was proved also for 8-aza-indolizine **60a** and its 7-oxo-analog **61b** [85]. 1-Azaindolizine bearing 2-SO_2_Me group failed to go in such cycloaddition [86], whereas the same structures with 2-SMe group **61a**,**b** [87] and their [h]-benzannelated derivatives **62** [88] formed the desired azacyclazines with DMAD. In our recent work, we proved that MAC could react with 1-azaindolizine **63** giving azacyclazine in the open air [36]. Diphenylacetylene was capable to transform imidazopyridine **64** to azacyclazine under the action of Pd(OAc)_2_/Cu(OAc)_2_ [89].

Mesoionic structure **65a** underwent cycloaddition with DMAD giving fully covalent structure **65b** proved by X-ray [55], Scheme 32.

Imidazopyridines **66** are transformed to azacyclazines under the action of Pd(OAc)_2_/Cu(OAc)_2_ [90] and [91], Scheme 33, Table 15.

Imidazo[1,2-a]pyridines and imidazo[1,2-a]pyrimidines readily reacted with diaryl acetylenes in presence of catalyst [92], Scheme 34, Table 16.

Separate catalyzed reaction of imidazopyrimidines **70** with diaryl acetylenes gave library of compounds with anti-inflammatory activity [11], Scheme 35, Table 17.

The new class of excited-state intramolecular proton transfer-capable molecules, benzo[a]cyclazines, bearing the 2-hydroxyphenyl substituent were prepared in a straightforward manner from imidazo[1,2-a]pyridines **71** via a tandem [8+2] cycloaddition–[2+6+2] dehydrogenation reaction using microwave [6], and similar reaction also involved imidazopyrimidine derivatives **72 [93]**, Scheme 36, Table 18.

A—1-TMS-2-OSO_2_CF_3_-benzene, CsF, 18-Crown-6, MW (25 min, 160 °C); B—1-TMS-2-OSO_2_CF_3_-benzene, CsF, 18-Crown-6, MW (90 W, 40 psi, 15 min, 80 °C); C—1-TMS-2-OSO_2_CF_3_-3-MeO-benzene, CsF, 18-Crown-6, MW (90 W, 50 psi, 15 min, 80 °C); D—1-TMS-2-OSO_2_CF_3_-4-MeO-benzene, CsF, 18-Crown-6, MW (90 W, 50 psi, 15 min, 80 °C)

A base promoted protocol for the synthesis of benzo[a]cyclazines from imidazopyridines and benzyne precursors under metal-free conditions was developed [94], Scheme 37, Table 19.

An interesting reaction that formally fit the [8+2] cycloaddition was developed for interaction of imidazopyridines **74** and 1,2-dihalobenzenes in presence of Pd-catalyst [95], Scheme 38. Table 20.

The system containing two fused imidazopyridines **75** was placed in reaction with DMAD [96], Scheme 39. One ring of imidazopyridine entered into [8+2] cycloaddition with the yields 22–30% on heating in benzene.

A rare example of benzonitrile entered into [8+2] cycloaddition to produce diazacyclazine **76b** was reported [97], Scheme 40. Azaindolizine **76a** reacted with BuLi giving dipolar structure **76c** which underwent cyclization.

## 8. Concerted One-Step 1,10 Processes

If one adds a multiple bond to the end of the tetraene fragment of indolizine, the ring closure becomes possible. A multiple bond can be alkene, alkyne or arene, and the “end” of the tetraene can be position 3 or 5. However, no such reactions exist for 3-vinyl/ethynyl derivatives and for 5-vinyl indolizines. The first example of such cyclization was reported for 5-ethynyl indolizine **77c** [98,99] which is postulated to be intermediate, Scheme 41.

According to [98] reaction **77a**–**77b** proceeded with a yield of 10–15%, later result [99] was 7%. The main product was 5-Me-3-benzoyl indolizine which could not be converted to **77b**. However, we showed that 5-ethynyl indolizine **77c** obtained by Sonogashira coupling [100] could not be converted to cyclazine **77b** under thermal or acidic conditions.

5-Iodo-indolizine **78a** in conditions of Sonogashira reaction with 2 eq of ethoxycarbonyl acetylene gave cyclazine **78b** [36], Scheme 42. We supposed that the reaction started from nucleophilic attack of acetylenide anion on **78c**.

5-Ethynyl derivatives of imidazopyridines **79a**–**c** behaved in an expected way [99], Scheme 43.

In one case the double bond of benzene ring at position 3 of indolizine **80a** underwent catalytic ring closure to benzocyclazine **80b** [35], Scheme 44.

A similar process was employed to obtain highly fluorescent benzo derivatives of azacyclazine starting from Br-substituted 3-aryl imidazopyridines **81**, Scheme 45, Table 20 [8,101].

We found that 5-chloro-3-benzoyl indolizines **82a**,**b** in acidic conditions closed the ring [102,103], Scheme 46, forming benzocyclazine derivatives **82c**,**d** (X = Cl 83%, X = NO_2_ 90%). Here the protonation opened direct link to 1,10-polyene which underwent ring closure.

## 9. Concurrence of [8+2] and [4+2] Cycloadditions

2-Styrylindolizine **83a** reacted with methyl acrylate (Scheme 47) giving the usual product of oxidative [8+2] cycloaddition—cyclazine **83b** together with [4+2] cycloadduct **83c** without catalyst [38]. After more prolonged heating (from 122 h to 288 h) the ratio **83b:83c** changed from 3:68 to 38:10. A somewhat similar result was obtained in reaction with N-ethylmaleimide where [4+2] adduct (33%) was formed together with isomeric dihydrocyclazines (43%).

Possibility of concurrence between [8+2] and [4+2] cycloaddition appeared in the case of 2-aryl substituted azaindolizines, Scheme 48. At least three papers appeared on this topic [7,104,105] and the data are summarized in Table 22.

## 10. Understanding the Mechanism: Michael Adducts, Hydrogenated Structures and Others

Reactions of [8+2] type of indolizines and their aza/benzo derivatives with acetylenes and alkenes are regioselective due to pronounced polarization of indolizine and (if any) of a multiple bond. Thus, the positive end of the double/triple bond (e.g., in E-C≡CH or in ECH=CH_2_) would be definitely attached to π-excessive pyrrole carbon C-3 without any exception, as is evident from all the tables. If the alkene/acetylene bears an electron-donating group and indolizine is appropriately polarized (e.g., by additional 6(8)-NO_2_ group), then regioselectivity is again preserved, and electronegative end of the multiple bond would be attached to π-deficient pyridine carbon C-5. 

### 10.1. Theory

There are theoretical quantum chemical calculations on [8+2] cycloaddition of alkenes to indolizines [106,107] with a variation of the polar nature of substituents in alkenes and comparing indolizine and 6-nitroindolizine. An ab initio and semiempirical (AM1 and SINDO1) calculations clearly confirm the possibility of three different mechanisms (Scheme 49). The concerted one-step mechanism (iii) is preferable, if there are no polar groups in a dienophile and indolizine. Another type of stepwise cycloaddition (electrophilic addition (i)—nucleophilic ring closure (ii)) should be realized for the case of nitroethylene. The last type of dipolar cycloaddition (nucleophilic addition (iv)—electrophilic ring closure (v)) would be expected for the reaction of 6-nitroindolizine with aminoethylene, Table 23.

However, indolizines (even activated by 6- or 8-NO_2_-group) failed to react with enamines or enols [107], although reaction with dialkylaminoacetylene is possible, Scheme 50. Although the 1:1 adduct was definitely not the product of [8+2] cycloaddition **87a,** rather it was [4+2] adduct of acetylene across the nitroethylene **87b**, its structure confirmed the regioselectivity of attack of aminnoacetylene to the position C-5 of indolizine.

After the addition of alkyne to position C-3 of indolizine, the initially formed zwitter-ion **88a** could be transformed to a covalent structure either forming the cycloadduct **88b** (i.e., dihydrocyclazine) or underwent shift of H-3 from acidic position C-3 to vinyl anion thus forming 3-vynyl derivative **88c**. Scheme 51.

### 10.2. 3-Vinyl Derivatives

In few cases, 3-vinyl substituted intermediates were isolated and characterized from reactions of indolizines and acetylenes, Scheme 52, Table 24. In the first experiment of reaction of indolizines with DMAD without any catalyst, the cis- and trans-adducts **89** were formed [108]. Cis- and trans-derivatives of pyrrolopyrimidone **90** and DMAD did not undergo further cyclization to azacyclazine in presence of Pd-C [85]. 1.8-Annelatyed indolizines gave purple 3-vinyl adducts **91** with DBZA [46] which underwent further dehydrogenation without cyclization (see Scheme 9). Benzoindolizines **92** [29] and their aza-derivative **93** [88] even in presence of catalysis gave the adducts together with cyclazines. 2-Isopropenyl indolizine **94** after prolonged heating with DMAD gave the mixture of isomeric 3-vinyl derivatives [109].

### 10.3. Dihydrocyclazines

First, dihydrocyclazine was obtained by Boekelheide [23] with a yield of 15% together with cyclazine. He tried to prove the position of protons by chemical tools and finally assigned the protons to be located as in **95** (Scheme 53), i.e., far from the attached DMAD. In 1984 Japanese chemists tried to prove the structure of all intermediated in the reaction of indolizines with DMAD in the absence of catalyst [108]. They proved two types of structures **96a** and **96b** (together with 3-vinyl adducts **89**) obtained with the yields 4–27% for **96a** and 5–6% for **96b**. Bis-(indolizinyl)etane formed the bis-dihydrocyclazine derivative **97** with a yield of 26% [39]. Azaindolizinone reacted with DMAD in presence of Pd-C giving about 4% of dihydro-compound **98** [85]. 

The structure of dihydrocyclazine depends on the nature of substituents in the ring. Thus, in our early work [110] we found that 6-nitroindolizine reacted with DMAD (PhMe/Δ/3h) giving the expected nitrocyclazine **99a** (Scheme 54) together with the cyclazine **99b** without NO_2_ group (31%:7%), which is formed presumably by elimination of HNO_2_ from dihydrocyclazine **99c**. 

In the paper [111] it was shown that 5-Me-indolizine derivative under the action of DMAD (PhH, rt) gave dihydrocyclazine **100a** with a yield of 54%, Scheme 55. Further reaction with the excess of DMAD give the macrocyclic cyclazine derivative **100b** [111] with the structure proved by X-ray, and it was not the structure of the 1:2 adduct (**100c**) postulated in [108].

The reaction of Mes_2_B-substituted acetylene with benzoindolizine at rt gave dihydrocyclazine **101** with 90% yield [37] (Scheme 56) which can be further aromatized. The same structure underwent cycloaddition with hetearyl acetylenes [66] (CH_2_ClCH_2_Cl/Δ/6 h) giving another type of dihydrocyclazines **102** (R = 2-pyridyl, 54% and R = 2-quinoline, 42%) which were converted to benzocyclazines under the action of sulfur (PhCl/Δ/10 h with the yields 59% and 42%). 3-CN substituted benzo[a]indolizine with DMAD (Pd-C/PhMe/Δ/2 h) gave 7% of the adduct of the structure **103** (together with benzocyclazine) and with di-t-BuOCO-acetylene the yield of cycloadduct is higher (42%) [65]. 

### 10.4. Alkenes

The reaction of indolizines with alkenes has attracted a lot of attention. The following potential dienophiles were used as 2π-components for potential [8+2] cycloaddition: nitroolefins, acrylonitrile, benzoquinone, methylvinylketone, alkyl acrylates, alkyl maleate, alkyl fumarate, maleic acid, maleic anhydride, N-substituted maleimide, 4-substituted-1,2,4-triazoline-3,5-dione, dialkyl azodicarboxylates, nitrile oxide, 1.2-dicyanocyclobutene and some other [38,53,112,113,114,115].

In most reactions, two types of products are observed: first from proton shifts in an intermediate zwitter-ion leading ultimately to the isolated Michael addition product at the position 3 of the indolizine or, second, deriving from hydrogen loss or shifts in the primary adduct giving [2+8] cycloadducts of tetrahydro-, dihydro- or (in rarest cases) aromatic cyclazines. 

In particular, indolizines reacted with maleates and acrylates giving [8+2] cycloadducts with the subsequent 1,5-hydrogen shift as in **104a**, Scheme 57 [112]. In most other cases Michael adducts at C-3 **104b** were formed. Benzo[a]indolizines with some dipolarophiles produced kinetically controlled cycloadducts **105a** which isomerized to Michael adducts **105b** [113]. For further discussion on the mechanism see ref. [116].

## 11. Conclusions

As is evident from all the schemes and tables, [8+2] cycloaddition of indolizines, their aza- and benzo derivatives leading to (aza/benzo) cyclazines is a big portion of modern organic chemistry, its concrete and powerful tool with its own achievements and secrets. There are a lot of catalyst and oxidants proposed to make the final aromatic structure, starting from oxygen, sulfur, Pd-C, Pd(OAc)_2_ and Pd complexes, Cu(OAc)_2_, MnO_2_, quinones (DDQ, benzoquinone), new tools appeared to stimulate reaction (blue LED, microwaves, etc.). The dependence of the process on the nature of substituents in the benzo/aza-substituted indolizines and alkynes/alkenes, the intermediacy of open chains cyclic derivatives made clearer the entire mechanism. Even 60 years after its first discovery, [8+2] cycloadditions continue to play an important part in organic synthesis.

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
