# Peer review of "Indolizines and Their Hetero/Benzo Derivatives in Reactions of [8+2] Cycloaddition"

_molecules, 2021, doi:10.3390/molecules26072050_

Round 1

Reviewer 1 Report

In this paper, the authors extensively reviewed some the indolizine chemistries and more specifically the use of indolizine derivatives in [8+2] cycloadditions. This review will add a significant contribution to the field. Therefore, I recommend acceptance after minor revisions:

  1. I t will be useful if the authors could add a paragraph in the introduction section discussing the applications of indolizine and its derivatives.
  2. Section 6 title: Change "were" to "where" (Cycloadditions were indolizines are intermediates.)

Author Response

1.    I t will be useful if the authors could add a paragraph in the introduction section discussing the applications of indolizine and its derivatives.
  - A phrase is added. Since indolizines are starting materials to cyclazines, we concentrated on the use of cyclazines [Ref. 2-11]
2.    Section 6 title: Change "were" to "where" (Cycloadditions were indolizines are intermediates.)
    - Done

Reviewer 2 Report

+ Abstract and the conclusion should be substantially extended. Key issues and conclusions should be included.

+ Cycloaddition reactions can be classified as (4n)-pi-electron processes (which under non-catalytic conditions are generally realised via stepwise mechanisms, or, (4n+2)-pi-electron (which under non-catalytic conditions are generally realised via one-step mechanisms. In some cases, however, deviations from these rules are possible. For example, some 6-pi-electron cycloadditions are realized via polar, stepwise mechanisms (Organics 2020, 1, 49-69). So, these informations as well as mechanisctic backgrounds for [8+2] cycloaddition processes should be included to the introduction.

+ Nature of catalysts interatcion with addents should be precised. This is activation of one or both components? What types of activation are possible in which cases? 

+ Scheme 12:
It is not possible, to obtain the adduct directly from 28 and R-substituted ethenes. In the course of cycloaddition, the hybridization on the carbon atom linked with the R substituent changes to sp3. There is an error in the structure, or at the scheme is a secondary reaction product rather than a cycloaddition product. I have similar remark regarding to the Scheme 13.

+ According to the actual state of knowledge (Molecules 2016, 21, 1319) term "concerted" is full-unjustified. The vast majority of single-step mechanisms with cyclic transitions are not "concerted" processes. So, term "concerted" should be replaced to the "one-step".

Author Response

+ According to the actual state of knowledge (Molecules 2016, 21, 1319) term "concerted" is full-unjustified. The vast majority of single-step mechanisms with cyclic transitions are not "concerted" processes. So, term "concerted" should be replaced to the "one-step".
   - on p.2: "concerted" is changed to "one-step (concerted)"
    on p.26 "Concerted 1,10 processes" is changed to "Concerted (one-step) 1,10 processes" 
    on p. 31 "The concerted mechanism"  is changed to "The concerted one-step mechanism"

+ Abstract and the conclusion should be substantially extended. Key issues and conclusions should be included.
   - The conclusion is expanded. To abstracts: we added "one- or two-steps mechanism". 

+ Cycloaddition reactions can be classified as (4n)-pi-electron processes (which under non-catalytic conditions are generally realised via stepwise mechanisms, or, (4n+2)-pi-electron (which under non-catalytic conditions are generally realised via one-step mechanisms. In some cases, however, deviations from these rules are possible. For example, some 6-pi-electron cycloadditions are realized via polar, stepwise mechanisms (Organics 2020, 1, 49-69). So, these informations as well as mechanisctic backgrounds for [8+2] cycloaddition processes should be included to the introduction.
   - We thank the refferee for the ref. (Organics 2020, 1, 49-69) which is devoted to 32CA reaction and to many examples of Huisgen cycloaddition, except the [8+2] reaction discussed in our review. We have added this important reference to our introduction. We do not want to initiate any discussion, since our early works (ref. 105 and 106) on ab initio and semiempirical calculations of [8+2] exist.

+ Nature of catalysts interatcion with addents should be precised. This is activation of one or both components? What types of activation are possible in which cases? 
  - There are no data on this topic in ALL papers throughout the review. We do not want to make any speculations.

+ Scheme 12: It is not possible, to obtain the adduct directly from 28 and R-substituted ethenes. In the course of cycloaddition, the hybridization on the carbon atom linked with the R substituent changes to sp3. There is an error in the structure, or at the scheme is a secondary reaction product rather than a cycloaddition product. I have similar remark regarding to the Scheme 13.
    - One way to achieve this process is to imagine electrophilic addition—nucleophilic ring closure (as on Scheme 49). However, we understand the refferee doubt and add at p.8 (end) "via cascade reaction" to Scheme 12 and on p.9 "via similar cascade reaction" to Scheme 13